# miR-148b as a Potential Biomarker for IgA Nephropathy

Santosh Kumar [1,*] [ID], C. Priscilla [1], Sreejith Parameswaran [2], Deepak Gopal Shewade [3] [ID],
Pragasam Viswanathan [4] [ID] and Rajesh Nachiappa Ganesh [1,*] [ID]

1 Department of Pathology, Jawaharlal Institute of Postgraduate Medical Education and Research,
  Puducherry 605006, India
2 Department of Nephrology, Jawaharlal Institute of Postgraduate Medical Education and Research,
  Puducherry 605006, India
3 Department of Pharmacology, Jawaharlal Institute of Postgraduate Medical Education and Research,
  Puducherry 605006, India
4 Department of Bio Sciences, School of Bio Sciences and Technology, Vellore Institute of Technology,
  Vellore 632014, India
* Correspondence: san0016@gmail.com (S.K.); drngrajesh@gmail.com (R.N.G.)

**Abstract:** Background: IgA nephropathy (IgAN) is one of the most common glomerular diseases worldwide. Approximately 25 percent of IgAN patients reach the kidney failure stage within twenty years of diagnosis. The histopathological examination of kidney biopsy is needed to confirm the diagnosis of IgAN. microRNA (miRNA) is a small RNA that plays an important role at the post-transcriptional level by downregulating mRNAs (messenger RNA). We tried to establish a miRNA-based biomarker for IgAN. Methods: We recruited 30 IgAN patients and 15 healthy controls as study participants after taking their informed written consent. A real-time PCR-based method was used for the absolute quantification of miRNAs. A logistic regression method and receiver operating characteristic analysis were performed to find the diagnostic and prognostic accuracy of miR-148b and let-7b for IgAN in histopathological MEST-C scores. Results: miR-148b and let-7b levels were higher in IgAN patients compared to the healthy controls. miR-148b was positively correlated with glomerular filtration rate (GFR) and negatively correlated with segmental glomerulosclerosis, tubular atrophy/interstitial fibrosis (T), and blood pressure (BP). The sensitivity, specificity, and area under the curve (AUC) of the receiver operating characteristic (ROC) for miR-148b against T were 0.87, 0.77, and 0.85, respectively. The threshold value of the miR-148b copy number was 8479 to differentiate the severe condition of IgAN. Conclusion: miR-148b can be used as a potential biomarker for IgAN.

**Keywords:** IgAN; biomarker; MEST-C; microRNA; miR-148b; let-7b; miR-155; tubular atrophy

## 1. Introduction

IgA nephropathy (IgAN) is a slowly progressive kidney disease whose pathogenesis is not completely known [1,2]. It is one of the most common glomerular diseases worldwide, where around 25 percent of IgAN patients reach the kidney failure stage within twenty years of diagnosis [3]. It is diagnosed by histopathological examination of a kidney biopsy with dominant or codominant deposition of IgA molecules at the mesangial region [4]. The IgA molecules deposited at the mesangial area were found of type IgA1 and galactose deficient [5].

A kidney biopsy is an invasive procedure and has its own complications. It is not usually repeated as well. A delayed kidney biopsy is another drawback where the kidney occurs significant damage by the time IgAN is diagnosed. The characteristics of IgAN and kidney biopsy practice seem to be different in different parts of the world, and it is not yet clear that it is the same disease worldwide [6–9]. The IgA nephropathy classification group in 2017 proposed a modified histopathological scoring system known as the Oxford MEST-C score (M—mesangial hypercellularity, E—endocapillary hypercellularity, S—segmental glomerulosclerosis, T—tubular atrophy/Interstitial fibrosis, C—crescents) [10]. MEST-C

score with cross-sectional clinical data at biopsy has shortened the time frame to accurately predict patients at risk of adverse kidney outcomes [11]. Among all histologic parameters of MEST-C, T and C can hold the accurate predictive value for the adverse outcome of the disease [12]. T0 reflects the absence, T1 reflects 25% to 50%, and T2 reflects 50% onwards of tubular atrophy/interstitial fibrosis involvement. In the same way, C0 represents no crescent, C1 reflects up to 25%, and C2 reflects crescents in over 25% of glomeruli.

microRNAs (miRNAs) are small and approximately 22 nucleotides in length [13]. They are highly conserved, found in the intronic region, and downregulate the mRNA post-transcriptionally [14]. We have selected miR-148b and let-7b as the proposed predictive markers based on a microarray-based dataset and published literature [15,16]. miR-148b and let-7b have been associated with the aberrant glycosylation process and found regulating the enzyme core 1 β 1,3 galactosyltransferase 1 (C1GALT1) and the enzyme N-acetylgalactosaminyltransferase 2 (GALNT2) respectively [17,18]. The aberrant glycosylation process in the hinge region of the IgA1 molecule has been associated with the pathogenesis of IgAN [19]. In a previous study, it was shown that the antiglycosylation process changed the miR-148b and let-7b expressions in IgAN patients [18]. It is considered that miR-148b and let-7b might play a role in the pathogenesis of IgAN and have the potential to be used as a diagnostic and prognostic marker for the prediction of IgAN. In this study, we have tried to find the diagnostic and prognostic importance of miR-148b and let-7b for IgAN.

## 2. Materials and Methods

### 2.1. Study Participant Selection

All consecutive native biopsies, reported as primary IgA nephropathy, were considered for the study, while secondary causes of IgA nephropathy, patients with systemic lupus erythematosus (SLE), Henoch Schonlein purpura (HSP), other autoimmune diseases, kidney carcinoma, human immunodeficiency virus infection, hepatitis, diabetes mellitus, and infection-related glomerulonephritis were excluded. The participants were in the age group of 15 to 70 years. Institute Human Ethics Committee approval was taken for the study. We recruited 30 IgAN patients and 15 healthy persons (age and sex-matched) as study participants after taking their informed written consent following the inclusion and exclusion criteria.

### 2.2. Study Population Description

The study was conducted at a tertiary healthcare research center in the coastal region in the southern part of India. The study participants had mixed food habits, and a few of the male participants used to drink alcohol occasionally. There was no history of kidney diseases in the family of the study participants. All the study participants were under corticosteroid therapy at the time of the sample collection for the microRNA expression experiment. Serum creatinine, GFR, blood albumin, and urine protein were recorded at the time of biopsy. The average serum creatinine, GFR, blood albumin, urine protein, and body mass index (BMI) values were 1.75 (1.05–3.55) mg/dL, 41 (19.5–89) mL/min per 1.73 m$^2$, 3.31 (0.87) g/dL, 1 (1–2.75) and 23.05 (3.78) kg/m$^2$, respectively. The IgAN patients were not under the corticosteroid therapy at the time of biopsy and whoever had a history of corticosteroid therapy was excluded from the study. The screening of the disease condition and the patient recruitment for the study were conducted at the nephrology department. The department of nephrology follows the KDIGO guidelines and is always updated to its recent advancements. For this study, KDIGO clinical practice guideline for glomerular diseases were also followed [20]. Few IgAN patients require corticosteroid therapy, and this was reflected in our study participants. Corticosteroid therapy influences the biochemical profile of the IgAN patients, but does not affect the microRNA expressions [18]. We used this knowledge, and serum creatinine, GFR, blood albumin, and urine protein values were recorded at the time of biopsy and before the corticosteroid therapy, while we considered the microRNA expression levels from only

those IgAN patients who were under corticosteroid therapy. Our study design reduced the number of IgAN patients to a serious limitation, but it ensured a homogeneity and reproducibility of the experiment.

### 2.3. Sample Collection for the microRNA Experiment

A total of 5 mL of blood was collected into EDTA tubes after taking the informed written consent from the study participants. The sample was processed within one hour of collection. Plasma was collected after centrifugation at $3000\times g$ for five minutes at a temperature of four degrees. Plasma was stored immediately in aliquots at $-80\ ^\circ$C until further processing.

### 2.4. microRNA Quantification

miRNA quantification was performed in four experimental stages. The first stage was performed as miRNA isolation from the plasma sample. The second step was to check the quality of the isolated miRNA. Fluorometer (Qubit 3.0, ThermoFisher, Waltham, MA, USA) was used for this experiment. The third stage was performed as cDNA conversion from the isolated miRNA. The fourth and final stage was performed for the amplification of the target miRNA to see the expression. Real-time PCR (Quant Studio 3, ThermoFisher) instrument was used for cDNA conversion and miRNA expression. The chemicals, reagents, primers, and probes were bought from ThermoFisher and Helini Biomolecules (Chennai, India). Kit manufacturer's protocol was followed for the microRNA quality check (ThermoFisher—Qubit microRNA Assay Kit), microRNA isolation, cDNA conversion, and microRNA quantification (HELINI microRNA Real-Time PCR Kit). The forward and reverse primers used for miR-148b were TCAGTGCATCACAGAAC and GCGATCGGTAAGTACCTGA, respectively. The forward and reverse primers used for let-7b were CTATACTTCCTACT-GCCT and GCGATCGGTAAGTACCTGA, respectively. The probe used in the experiment was CGTACGTACGTA. All nucleotide sequences are in five to three prime directions. The mean expression value method was used for the normalization of miRNAs. The standard curve method was used for the absolute quantification and calculation of the copy number of miRNAs. We followed a previously established method for miRNA quantification in our laboratory using miR-155 as a reference control [21].

### 2.5. Histopathological Analysis

Kidney biopsy of IgAN patients was examined and interpreted through histopathological analysis through immunofluorescence light microscopy (Olympus BX51, Tokyo, Japan). MEST-C scoring under Oxford classification was used and kidney biopsy findings were documented.

### 2.6. Statistical Analysis

Shapiro–Wilk normality test was performed to check the data distribution. Wilcoxon's rank sum test and t-test were performed for probability distinction according to the data distribution. Logistic regression method was used for the predictive features of miRNAs, BMI, serum creatinine, GFR, albumin, blood pressure, and urine protein (dip stick method) against MEST-C score and CKD stages. All the possible statistical models were made by the above-mentioned variables. Area under curve (AUC) of receiver operating characteristic (ROC), sensitivity (SE), specificity (SP), Akaike information criteria (AIC) and Bayes information criteria (BIC) were used for the best fit model selection [22]. The cutoff value for AUC, SE, and SP was set at 0.70. All the models' validities were checked where SE, SP, and AUC were higher than 0.70. Correlation analysis was performed by Spearman's rank method. The cutoff value for the correlation coefficient was fixed at 0.33 at *p* value $< 0.05$. Mean values were given with standard deviation (SD) and median values with interquartile range (IQR). GFR remission with time (follow up) was estimated with the Kaplan–Meier method and comparison was made using the log-rank test. GFR remission was calculated for follow-up to 18 months against T0, T1, C0, and C1 (C2 stage considered

and counted under C1 category). The receiver operating characteristic curve was made by pROC package of R [23]. All the statistical analysis was performed using R version 4.2.2 (R Core Team, 2022, Vienna, Austria) [24].

## 3. Results

The mean age group of IgAN participants was 29 (9.63) years. About 47% of IgAN patients were male and 53% were female. The average systolic and diastolic blood pressure of the patients was 122 (110–142) and 82 (70–92.50) mm Hg, respectively, and was found to be in stage-1 hypertension state [25].

The copy numbers (median) of miR-148b in IgAN and healthy controls were 11,742 and 4032, respectively (Table 1). The copy numbers (median) of let-7b in IgAN and healthy controls were 1124 and 205, respectively (Table 1). The levels of miR-148b and let-7b in IgAN were 2.91 and 5.48 times higher than the healthy controls and were statistically significantly different, with $p$ values of 0.0002 and 0.0065, respectively. The miR-148b level was found higher in M1 and E1 than M0 and E0, respectively (Table 1). There was no statistically significant difference found in miR-148b levels in different stages of M and E. Furthermore, miR-148b was found decreased with the increase in S, T, C, and CKD stages (Table 1). miR-148b was found to statistically differentiate the S0 and S1, T0 and T1, C0 and C2, and CKD3 and CKD4 stages, with $p$ values of 0.031, 0.003, 0.006, and 0.012, respectively (Table 1).

**Table 1.** microRNA concentrations (copy number).

| Predictive | miR-148b Copy Number | $p$ | let-7b Copy Number | $p$ |
|---|---|---|---|---|
| Parameters | Median (IQR) | | Median (IQR) | |
| IgAN | 11,742 (5224–17,694) | 0.0002 | 1124 (202–2808) | 0.0065 |
| HC | 4032 (2970–5342) | | 205 (167–435) | |
| M0 | 7222 (3323–11,742) | 0.2661 | 174 (152–862) | 0.0624 |
| M1 | 13,973 (6104–18,083) | | 1990 (468–2900) | |
| E0 | 10,699 (5159–18,415) | 0.8716 | 1967 (519–2818) | 0.1558 |
| E1 | 12,784 (7745–17,146) | | 154 (150–166) | |
| S0 | 17,146 (7745–25,410) | 0.0313 | 707 (174–2712) | 0.5165 |
| S1 | 6788 (4255–13,658) | | 1999 (418–2958) | |
| T0 | 14,422 (9503–20,665) | 0.003 | 1978 (612–2837) | 0.0959 |
| T1 | 5072 (3520–7027) | | 296 (157–999) | |
| C0 | 14,191 (9503–17,694) | 0.8204 | 1978 (566–2808) | 0.7785 |
| C1 | 12,602 (5566–31,703) | | 628 (363–2607) | |
| C1 | 12,602 (5566–31,703) | 0.2403 | 628 (363–2607) | 0.5025 |
| C2 | 5202 (4222–7164) | | 292 (157–1934) | |
| CKD1 | 17,449 (13,154–19,165) | 0.9333 | 1564 (602–2943) | 0.2141 |
| CKD2 | 15,965 (13,110–20,012) | | 2888 (2605–3798) | |
| CKD2 | 15,965 (13,110–20,012) | 0.9143 | 2888 (2605–3798) | 0.019 |
| CKD3 | 12,404 (8402–30,708) | | 628 (279–1005) | |
| CKD3 | 12,404 (8402–30,708) | 0.0126 | 628 (279–1005) | 0.662 |
| CKD4 | 4112 (2694–5762) | | 336 (145–2946) | |
| CKD4 | 4112 (2694–5762) | 0.4606 | 336 (145–2946) | 0.8081 |
| CKD5 | 9321 (4255–13,658) | | 1999 (418–2958) | |

IgAN: IgA nephropathy; HC: healthy control, microRNA concentration is expressed in copy number/μL RNA (isolated RNA eluted in 20 μL nuclease-free water); $p$: $p$ value; IQR: interquartile range; M0, M1, E0, E1, S0, S1, T0, T1, C0, C1, and C2 are kidney biopsy interpretation scores of MEST-C Oxford scoring system for the IgAN; CKD1: chronic kidney disease stage one; CKD2: chronic kidney disease stage two; CKD3: chronic kidney disease stage three; CKD4: chronic kidney disease stage four; CKD5: chronic kidney disease stage five.

The let-7b level was found increased in M1 and S1 than MO and S0, respectively (Table 1). Furthermore, the level of let-7b was found to decrease with the increase in S, T, and C scores (Table 1). The CKD stage saw a zigzag at the let-7b level. There was a statistically significant difference found in let-7b levels between CKD stages 2 and 3, with a

*p*-value of 0.02. We could not find let-7b statistically differentiating any other predictive stages of IgAN.

Both the microRNAs (miR-148b and let-7b) showed a similar trend with tubular atrophy/interstitial fibrosis and crescents (Figure 1). miR-148b and let-7b kept decreasing with the increase from T0 to T1 (there was only one study participant who obtained a T2 score, and it was counted under T1) and from C0 to C1 to C2 (Figure 1). The microRNA level increased in IgAN (with respect to healthy group) and again started decreasing with the increase in severity of the disease. The microRNA level was still higher at the T1 and C2 stage than the HC, but the difference was not significant.

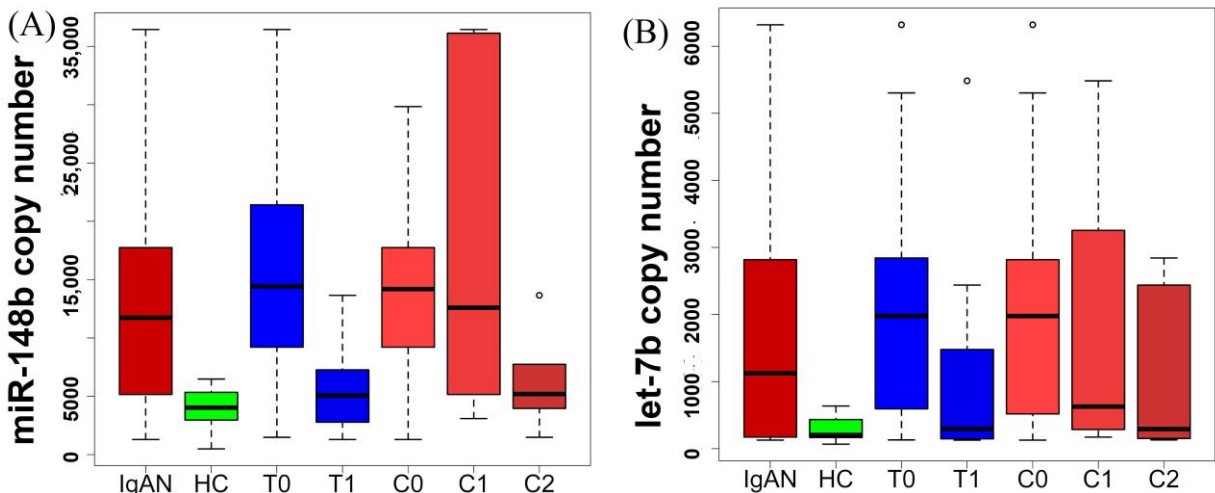

**Figure 1.** microRNA expression pattern with respect to histopathological stages in IgA nephropathy. IgAN: IgA nephropathy; HC: healthy control, microRNA concentration is expressed in copy number/μL RNA (isolated RNA eluted in 20 μL nuclease-free water); T0, T1, C0, C1, and C2 are kidney biopsy interpretation score of MEST-C Oxford scoring system for the IgAN. (**A**): miR-148b expression (copy number) with respect to histopathological stages in IgA nephropathy. (**B**): let-7b expression (copy number) with respect to histopathological stages in IgA nephropathy.

### 3.1. Correlation Analysis

We performed a correlation analysis of microRNAs with histopathological and biochemical predictive markers to find the association and the strength of the association. We found miR-148b in negative correlation with S, T, systolic, and diastolic blood pressure, and positively correlated with GFR based on Spearman's rank correlation analysis, where the correlation coefficient was set at more than ±0.33 and the *p*-value was less than 0.05 (Table 2).

**Table 2.** Correlation of microRNA with predictive markers of IgA nephropathy.

| microRNA | Predictive Marker | ρ | *p* |
|---|---|---|---|
| miR-148b | S | −0.400 | 0.028 |
| | T | −0.540 | 0.002 |
| | Systolic BP | −0.490 | 0.006 |
| | Diastolic BP | −0.420 | 0.019 |
| | GFR | 0.533 | 0.002 |
| let-7b | C3 | 0.360 | 0.047 |

ρ: correlation coefficient (Spearman's rank correlation); *p*: probability; S: segmental glomerulosclerosis; T: tubular atrophy/interstitial fibrosis; BP: blood pressure; GFR: glomerular filtration rate (chronic kidney disease epidemiology collaboration method); C3: complement component C3.

### 3.2. Logistic Regression Model

Based on logistic regression models, miR-148b was found to independently differentiate the tubular atrophy/interstitial fibrosis and crescent states in glomeruli. miR-148b with

GFR was found to be the best model to differentiate the tubular atrophy/interstitial fibrosis stages. However, GFR could not give any advantage to miR-148b (standalone marker) in terms of SE, SP, and AUC of ROC, which remained the same at 0.87, 0.77, and 0.85, respectively (Figure 2A). GFR as a standalone model also gave almost similar SE, SP, and AUC of ROC at 0.87, 0.72, and 0.86 to differentiate the tubular atrophy/interstitial fibrosis stages. miR-148b and GFR both have almost the same predictive capacity to differentiate T0 and T1, and hence the severity of IgAN, but together, they did not perform better as a marker. The ROC graph is the same for both the models, i.e., miR-148b with GFR and miR-148b alone (Figure 2A). The threshold values of miR-148b and GFR were found at copy number 8479 and 33 mL/min per 1.73 m$^2$, respectively. The threshold of miR-148b divided the IQR of T0 and T1, but this did not happen with the threshold of GFR (Figure 3). The GFR threshold merged in the IQR of T0 (Figure 3).

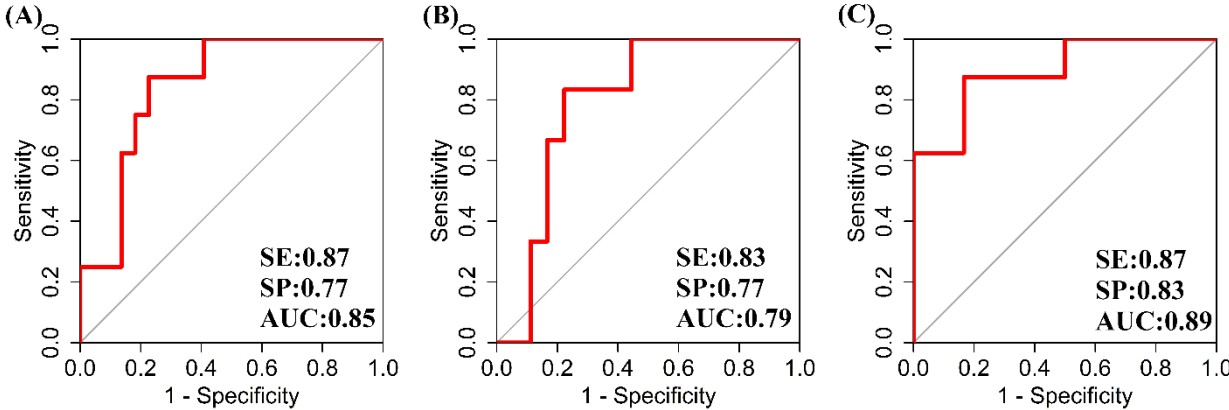

**Figure 2.** Predictive properties of miR-148b in IgA nephropathy. Figure legend: (**A**): receiver operating characteristic (ROC) curve of miR-148b with respect to T (tubular atrophy/interstitial fibrosis); (**B**): ROC of miR-148 with respect to C (crescents); (**C**): ROC of miR-148 with respect to chronic kidney disease (CKD) stages three and four.

None of the logistic regression models were able to significantly differentiate and predict the different states of crescents except for C0 and C2. GFR and urine protein together was found to be the best model to differentiate C0 and C2 with SE, SP, and AUC of ROC, at 1, 0.72, and 0.85, respectively. It was followed by urine protein, GFR, and miR-148b independently, differentiating C0 and C2, and hence the disease severity. The SE, SP, and AUC of ROC were found for urine protein, GFR, and miR-148b at 1, 0.78, and 0.89; 1, 0.72., and 0.88; and 0.83, 0.78, and 0.79 respectively. Although urine protein and GFR together was the most stable and best model predicted, urine protein and GFR independently were still found to have better SE and SP to differentiate the C0 and C2 stage or the disease severity. The threshold for urine protein, GFR, and miR-148b to differentiate the disease progression or the crescentic stages C0 and C2 were 1.5, 33 mL/min per 1.73 m$^2$, and copy number 8479, respectively. None of the three predictors' threshold values overlapped in the C0 or C2 region. Out of all the predictive markers, only miR-148b and GFR maintained the same threshold value of copy number 8479 and 33 mL/min per 1.73 m$^2$, respectively, in tubular atrophy/interstitial fibrosis and crescentic stages in IgAN.

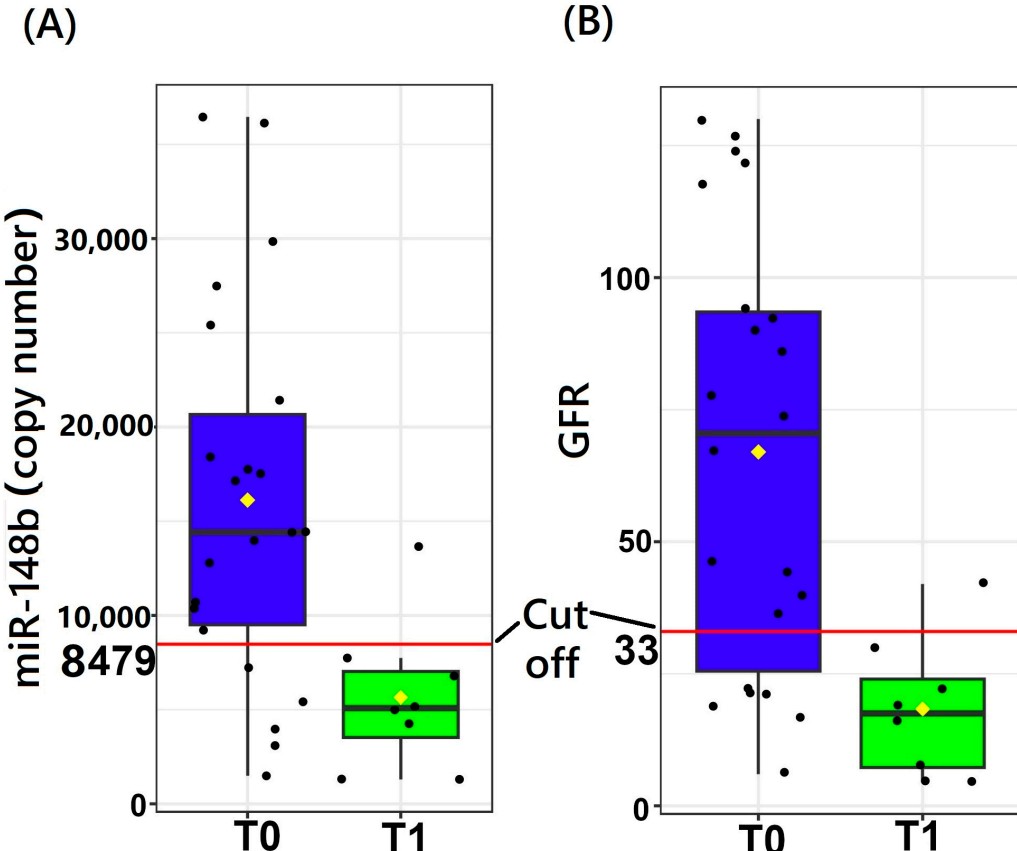

**Figure 3.** miR-148b and GFR in tubular atrophy/interstitial fibrosis states in IgAN. (**A**): miR-148b cutoff (threshold) level to differentiate IgAN severity based on T stage differentiation. (**B**): GFR cutoff (threshold) level to differentiate IgAN severity based on T stage differentiation. Threshold (cutoff) value estimated by a logistic regression model.

We used miRNAs to predict the CKD stages by the logistic regression model. The miR-148b showed good predictive value to differentiate the CKD stage 3 and 4 with SE, SP, and AUC of ROC at 0.87, 0.83, and 0.89, respectively (Figure 2C). The threshold value of miR-148b was found lower at copy number 7266 for the CKD stage. We could not find let-7b as a good predictive marker to differentiate the tubular atrophy/interstitial fibrosis, crescentic, and CKD stages in IgAN.

*3.3. GFR Remission in Follow Up*

We followed the GFR of study participants for 18 months. The GFR follow-up was categorized under T0, T1, C0, and C1. The C1 group comprised both C1 and C2. Study participants with kidney biopsy score T0 showed 62% remission of GFR at 18 months and those with T1 showed no remission at 16 months (Figure 4A). In the same way, study participants with kidney biopsy score C0 showed 67% remission of GFR and those with C1 showed 10% remission at 18 months (Figure 4B). There was a significant difference between the T0 and T1 groups with a *p* value of 0.011. C0 and C1 were not significantly different, with a *p* value of 0.128.

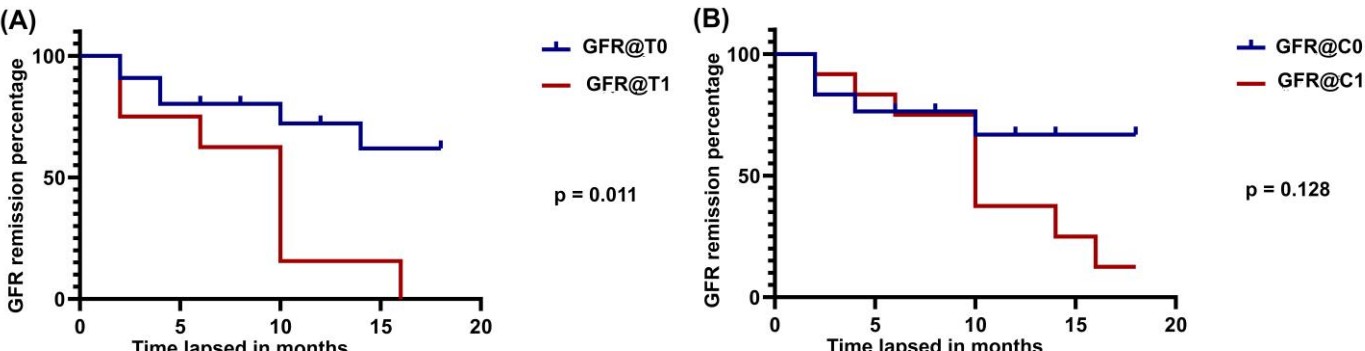

**Figure 4.** GFR remission in IgA nephropathy. (**A**): GFR remission curve in follow-up to 18 months with respect to the conditions against T0 and T1. GFR@T0—GFR remission in T0 stage. GFR@T1—GFR remission in T1 stage (T2 stage is also combined with T1). (**B**): GFR remission curve in follow-up to 18 months with respect to the conditions against C0 and C1. GFR@C0—GFR remission in C0 stage. GFR@C1—GFR remission in C1 stage (C2 stage is also combined with C1). Remission curve made with the Kaplan–Meier method and differentiation between the groups measured by log-rank test.

## 4. Discussion

G Serino reported that let-7b was upregulated and miR-148b downregulated in serum samples of IgAN than healthy controls [17]. They proposed let-7b together with miR-148b as markers for IgAN. In our study, miR-148b and let-7b were both upregulated by 2.91 and 5.48 times, respectively, compared to the healthy controls. If we compare our results with the previous study, which was a multicenter study spread over Italy, Greece, Japan, and China, we found a contradictory result. miR-148b level in plasma was almost three times higher than healthy participants. The level of let-7b was also over five times higher than the healthy controls, while in the previous study, it was around two to three times higher than the HC. There are differences between our study and the previous multicenter study. They worked on the serum sample while we used the plasma sample, and other major reasons were the larger sample size conducted in different ethnic groups. We do not think that serum or plasma would make a big impact on the result of the miRNA level. Here, the large sample size and ethnicity of the population certainly could have made a difference, as ours was a single-center study with a small sample size.

We furthermore studied the prognostic value of miR-148b and let-7b. Though let-7b was over five times higher in IgAN participants than the HC, it could still not differentiate any of the kidney biopsy histopathological stages, while miR-148b could differentiate the S and T stages irrespective of its lower level increase in IgAN than HC compared to let-7b. Tubular atrophy/interstitial fibrosis is considered as an independent parameter to prognose the disease severity [11]. At over 80% sensitivity and AUC, miR-148b alone could distinguish the different stages of T. miR-148b was negatively correlated to S, T, and blood pressure (BP) and positively correlated to GFR. We found an unusual behavior between microRNAs (miR-148b and let-7b) and IgAN severity. On average, both microRNAs' concentrations were higher in IgAN than HC, but with the increase in disease severity, the microRNA level was decreased. The average copy number of miR-148b in the T0 stage was 14,422, and in T1 it was 5072. The average copy number of miR-148b in healthy controls was found to be 4032. In the same way, the copy numbers of miR-148b in C0, C1, and C2 were 14,191, 12,602, and 5202, respectively, where the lower IQR of C2—4222—was close to the average copy number of miR-148b in healthy controls. The average miR-148b copy number (4112) in CKD stage four was almost the same as the average copy number (4032) of miR-148b in healthy controls. A similar trend to miR-148b was found in let-7b regarding the T and C stages. However, let-7b could not differentiate the different stages of T and C significantly. NM Kouri et al. in a recent study observed a decreased expression in miR-148b and let-7b in the presence of crescents in the Greek population [26]. This is the same group who previously reported miR-148b and let-7b importance in IgAN. If we

compare our study with them, then we have similarities and contradictions. Unlike us, they found let-7b to be a potential marker and not miR-148b, but the decreased level of microRNAs in the presence of C is a similar finding to ours. In a recent study, M Dhawaa et al. found miR-148b upregulated in IgAN by approximately four times and in system lupus erythrocytes (SLE) by approximately nine times [27]. Their findings also support our result that miR-148b has higher concentration in IgAN. Again, they showed a further increase in miR-148b for SLE. This way, miR-148b can distinguish/differentiate IgAN and SLE well, because miR-148b starts decreasing with the increase in disease severity, while the miR-148b level rose much higher in SLE. Therefore, miR-148b can be used as a disease severity marker for IgAN and a diagnostic marker for IgAN and SLE.

In our study, microRNAs have shown a bidirectional property as a marker. It is increased in preliminary diseased conditions and again starts decreasing with the disease severity. Both microRNAs have shown the same property in the T and C stages. Our result shows that unlike creatinine, GFR, urine protein, albumin, and other one-directional markers, miR-148b and let-7b are bidirectional. We strongly believe that the contradictory result found in previous studies are because of the bidirectional property of miR-148b and let-7b, which was not observed previously. In any study with a large cohort, a good proportion of T1 or T2 or/and C2 scores may cause the decrease in miR-148b and let-7b expression/concentration in IgAN. It might be the reason for miR-148b different (reduced) expression level in previous studies. This unique property of miR-148b and let-7b definitely makes them a potential biomarker for IgAN and needs further study to explore the changing behavior of miR-148b and let-7b in IgAN and other diseases.

The threshold of the miR-148b copy number to differentiate the T0 and T1 and C0 and C2 stages was the same, at 8479. The threshold of miR-148b to differentiate CKD stages three and four were found at a copy number of 7266. Therefore, we can fix a cutoff point for miR-148b to differentiate the disease severity. Copy number 8479 could be used as a disease severity mark for IgAN, whereas 7266 could be used as another deteriorating condition. Therefore, once IgAN is confirmed, decreased miR-148b could be used as a predictive tool for the progression of the disease.

Among conventional markers, GFR and urine protein showed high predictive values for disease progression. miR-148b and GFR showed the same threshold value in the T and C stages. The threshold value of GFR in the logistic regression model could not differentiate the IQR of T stages, unlike miR-148b, and hence miR-148b has an advantage over GFR as a better disease severity marker. To differentiate the crescentic stage, urine protein and GFR proved to be better markers than miR-148b.

In disease progression through to GFR follow-up, T1 was the worst histopathological stage reported, as there was no remission of GFR observed in a follow-up of more than a year. It shows that T is the best histopathological condition to predict disease severity, and thus supports the VALIGA group report, where they found that GFR decline was only associated with T in patients with immune-suppression therapy [28].

Although miR-148 has demonstrated good predictive properties as a biomarker for IgAN, it would still be premature to see it as a replacement of kidney biopsy. However, kidney biopsy is an invasive procedure, and the only confirmative method for the diagnosis of IgAN [29]. If kidney biopsy were not an invasive procedure, it would be the best method for the disease prognosis as well. Unfortunately, the main constraint of repeat biopsy is its invasiveness and complication. That is the reason why we need a noninvasive biomarker specifically for IgAN. miR-148b serves a purpose and can be a useful tool for predicting the disease severity by filling the gap created by the lack of or need for repeat biopsy. Our study has limitations of sample size and a lack of other glomerular diseases as control, but these limitations do not diminish miR-148b as a potential biomarker for IgAN.

## 5. Conclusions

miR-148b and GFR best predict IgAN severity, and miR-148b has an advantage over GFR because of its unique bidirectional property. miR-148b differentiates the severity of

IgAN with a threshold value, and its decreasing concentration indicates the progression of the disease. As repeat biopsy is not performed frequently, miR-148b can be used as an alternative biomarker for disease progression. It can also be considered as a prospective diagnostic tool for the detection of IgAN among patients with reported kidney dysfunction at an early stage.

**Author Contributions:** Conceptualization, S.K., S.P., P.V. and R.N.G.; methodology, S.K., C.P., D.G.S. and R.N.G.; software, S.K.; validation, S.K., C.P., S.P., D.G.S., P.V. and R.N.G.; formal analysis, S.K. and R.N.G.; investigation, S.K., S.P. and R.N.G.; resources, S.K., S.P., D.G.S. and R.N.G.; data curation, S.K., C.P. and R.N.G.; writing—original draft preparation, S.K.; writing—review and editing, S.K., C.P., S.P., D.G.S., P.V. and R.N.G.; visualization, S.K. and R.N.G.; supervision, R.N.G.; project administration, R.N.G.; funding acquisition, S.K. and R.N.G. All authors have read and agreed to the published version of the manuscript.

**Funding:** This research was funded by Science and Engineering Research Board (SERB), Government of India, grant number—EMR/2016/003382 and Jawaharlal Institute of Postgraduate Medical Education and Research, Puducherry—JIP/Path/IMRP/SanKum/2015.

**Institutional Review Board Statement:** The study was conducted in accordance with the Declaration of Helsinki, and approved by the Institutional Review Board (or Ethics Committee) of Jawaharlal Institute of Postgraduate Medical Education and Research, Puducherry (JIP/IEC/SC/2015/19/785) for studies involving humans.

**Informed Consent Statement:** Informed consent was obtained from all subjects involved in the study.

**Data Availability Statement:** The data presented in this study are available on request from the corresponding author. The data are not publicly available due to ethical constraints related to patient information details.

**Conflicts of Interest:** The authors declare no conflict of interest.

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
