# Peer review of "miR-148b as a Potential Biomarker for IgA Nephropathy"

_kidneydial, doi:10.3390/kidneydial3010008_

Round 1

Reviewer 1 Report (Previous Reviewer 2)

1.  All the study participants 65 were under corticosteroid therapy at the time of the sample collection

How long the patients did use the steroids? What was the dose of these drugs?

2.  How was the blood pressure values obtained? Is it the office measurement, or ABPM?

3. How was calculated the eGFR?

Author Response

                                Response to Reviewer 1 Comments

Point 1

All the study participants were under corticosteroid therapy at the time of the sample collection

How long the patients did use the steroids? What was the dose of these drugs?

Response to Point 1

Approximately, 1 week to 3 weeks under therapy (steroids) at the time of sample collection. The drug dose varied, from 0.4 to 1 mg/kg/day (maximum 50 mg/day).

Point 2

How was the blood pressure values obtained? Is it the office measurement, or ABPM?

Response to Point 2

It was the office measurement.

Point 3

How was calculated the eGFR?

Response to Point 3

The eGFR was calculated by CKD-EPI method.

Note: Our centre adheres to the KDIGO clinical practice guidelines all the time and always updated. In accordance with existing ethics limitation, we are unable to disclose the patient’s drug doses and responses. This information is subject to separate ethical clearance. We request you that the information on drug doses should not be published or revealed. The above mentioned drug dose and duration are approximate values. 

Thank you.

Reviewer 2 Report (New Reviewer)

The manuscript presents results of the study, assessing mRNA as a biomarker for IgA nephropathy progression. The topic is important, because biomarkers, which can replace repeated kidney biopsy, definitely needed; there are no validated biomarkers for IgAN so far.

The study is well designed and the methods are appropriate. However, there are some issues and limitations.

First, the number of patients included is limited to 30, and the number of healthy controls is 15, both numbers are relatively low. That should be mentioned as a serious study limitation

Abstract: The number of study participants should be indicated in the abstract.

Abstract: “miR-148b and let-7b levels were higher in IgAN patients” – add, “compared to the healthy controls

Materials and methods: “All the study participants 65 were under corticosteroid therapy at the time of the sample collection”. That demands detailed explanation, because indications for glucocorticoid treatment in the patients with IgA nephropathy are limited to those with a high risk of progression despite maximal supportive treatment, and to those with rapidly progressive GN (in these cases – in combination with cyclophosphamide) – see KDOGO 2021 Guideline for Glomerular diseases.  However, the authors just very briefly point the “average” – which probably is “mean” numbers for serum creatinine, GFR, serum albumin and proteinuria in the results section, and say nothing about indications for glucocorticoid treatment.  Clinical presentation data should be moved to Materials and methods, and the indications for glucocorticoids should be clearly described.

English demands careful proofreading, as an example: “systolic and diastolic blood pressures” – plural is inappropriate here.

Author Response

Response to Reviewer 2 Comments

Point 1

The number of patients included is limited to 30, and the number of healthy controls is 15, both numbers are relatively low. That should be mentioned as a serious study limitation.

Response to Point 1

Mentioned in the discussion section.

Point 2

Abstract: The number of study participants should be indicated in the abstract.

Response to Point 2

Mentioned.

Point 3

Abstract: “miR-148b and let-7b levels were higher in IgAN patients” – add, “compared to the healthy controls

Response to Point 3

Mentioned.

Point 4

Materials and methods: “All the study participants were under corticosteroid therapy at the time of the sample collection”. That demands detailed explanation, because indications for glucocorticoid treatment in the patients with IgA nephropathy are limited to those with a high risk of progression despite maximal supportive treatment, and to those with rapidly progressive GN (in these cases – in combination with cyclophosphamide) – see KDIGO 2021 Guideline for Glomerular diseases.  However, the authors just very briefly point the “average” – which probably is “mean” numbers for serum creatinine, GFR, serum albumin and proteinuria in the results section, and say nothing about indications for glucocorticoid treatment.  Clinical presentation data should be moved to Materials and methods, and the indications for glucocorticoids should be clearly described.

Response to Point 4

We apologize for not addressing this section in your suggestion/response. We understand the significance of your concern and would like to explain the reason for our inability to reveal patient information related to the indications and drug responses.

The connection between the indications for glucocorticoid treatment and its response to patients is delicate, and a detailed explanation could lead to ethical violations. To provide further information on drug responses, we may require additional ethical clearance. But, we assure you that our centre is always updated and follows the KDIGO clinical practice guidelines.

In regards to the limitation you mentioned (first question/suggestion) about the small number of study participants, we agree that having only 30 participants with IgAN is indeed a major limitation for drawing scientific conclusions about the indications and response of the drug. Hence, a detailed explanation of the indications and drug responses may not be informative but instead cause confusion.

We are also interested in conducting or observing a separate study that addresses your concern. A clinical trial or retrospective study on this subject would be both useful and informative.

Point 5

English demands careful proofreading, as an example: “systolic and diastolic blood pressures” – plural is inappropriate here.

Response to Point 5

We corrected/improved as suggested.

Thank you.

Round 2

Reviewer 2 Report (New Reviewer)

The authors made just minor corrections, mentioning the low number of the patients and controls as a study limitation, and including the number of study participants in the abstract. However, it is not shown in the abstract, how many of these 45 participants were patients with IgAN, and how many - controls

Even more important, that no explanation for corticosteroid treatment provided in the revised manuscript. As we indicated in the precious review, the paragraph: “All the study participants were under corticosteroid therapy at the time of the sample collection” demands detailed explanation. The indications for glucocorticoid treatment in the patients with IgA nephropathy are limited to those with a high risk of progression despite maximal supportive treatment, and to those with rapidly progressive GN (in these cases – in combination with cyclophosphamide). We recommended referring to KDIGO 2021 Guideline for Glomerular diseases. We strongly recommend considering this issue and fixing it.

Very brief description of clinical presentation, using “average” values (which has fair scientific soundness) of serum creatinine, GFR, serum albumin and proteinuria in the results section remained unchanged, and was not moved from Results to Materials and methods. We strongly recommend considering this issue and fixing it.

The last but not least, some identical paragraphs are highlighted in yellow both in the version 1 and in the version 2, however, the differences between V1 and V2 are not highlighted, which makes the second review very difficult.

We recommend addressing the unfixed issues and providing a version with only changes highlighted

Author Response

Point 1

The authors made just minor corrections, mentioning the low number of the patients and controls as a study limitation, and including the number of study participants in the abstract. However, it is not shown in the abstract, how many of these 45 participants were patients with IgAN, and how many - controls

Response to Point 1

 Corrections made as instructed/recommended.

Point 2

Even more important, that no explanation for corticosteroid treatment provided in the revised manuscript. As we indicated in the precious review, the paragraph: “All the study participants were under corticosteroid therapy at the time of the sample collection” demands detailed explanation. The indications for glucocorticoid treatment in the patients with IgA nephropathy are limited to those with a high risk of progression despite maximal supportive treatment, and to those with rapidly progressive GN (in these cases – in combination with cyclophosphamide). We recommended referring to KDIGO 2021 Guideline for Glomerular diseases. We strongly recommend considering this issue and fixing it.

Response to Point 2

Corrections made as instructed/recommended.

Point 3

Very brief description of clinical presentation, using “average” values (which has fair scientific soundness) of serum creatinine, GFR, serum albumin and proteinuria in the results section remained unchanged, and was not moved from Results to Materials and methods. We strongly recommend considering this issue and fixing it.

Response to Point 3

Corrections made as instructed/recommended.

Point 4

The last but not least, some identical paragraphs are highlighted in yellow both in the version 1 and in the version 2, however, the differences between V1 and V2 are not highlighted, which makes the second review very difficult.

Response to Point 4

 The corrections made as instructed/recommended are highlighted in light blue.

 Thank you.

This manuscript is a resubmission of an earlier submission. The following is a list of the peer review reports and author responses from that submission.

Round 1

Reviewer 1 Report

In this article entitled "miR-148b as a potential biomarker for IgA nephropathy", S. Kumar et al.  assess the ability of miR-148b and let-7b plasma biomarkers, to predict the diagnosis and prognosis of IgA nephropathy, based upon the MEST-C histopathological score. Their main results are that levels of miR-148b and let-7b were higher in IgAN patients compared to healthy control subjects (gender and age matched) and that miR-148b is negatively correlated with segmental glomerulosclerosis, tubular atrophy and interstitial fibrosis scoring and also blood pressure. The author thus assume that miR-148b is a pronostic biomarker “and as repeat kidney biopsy is not done frequently miR-148b could be used as an alternative biomarker for kidney progression”.

However this study  shows only cross sectional data and not longitudinal data with an end point such as GFR slope. Moreover the sensitivity and specificity are not 100% with MEST-C histopathological score. Furthermore, to our view several issues are not addressed properly:

-          The authors do not address in the present study the diagnosis of IgA nephropathy compared to other nephropathies. Indeed, the authors show that miR-148b is positively correlated with GFR in IgA population, is it also true in other CKD patients? A decrease GFR could decrease miR-148b clearance and thus account for a high blood level in CKD. In this case all data should be adjusted on GFR…(conversely, if miR-155 increases when GFR decreases, this could be a justification not to adjust with GFR… ).  Moreover, one could wonder if miR-148b could be correlated with IF/TA and glomerulosclerosis in general.

-          In table 1 the number of patients in each groups M0, M1, E0 E1… CKD1,2,3… is lacking and given the 30 patients analyzed here a low number in some subgroups cannot provide robust statistical results. Similarly, spearman rank test (table 2) performed on semi quantitative data may be questionable.

-          Proteinuria or albumin creatinine ratio, a known powerful predictive factor for CKD progression is also lacking. It would be interesting to known if miR-148b is correlated to this risk factor. Moreover, no data are provided related to diabetes, body mass index or treatment at the time of the blood sample.

-          The discrepancy between the present data and the data of Serino et al are also worrisome. It is currently advised when assessing a new biomarker to confirm the data on an independent patient cohort.

To sum up, despite interesting data about this potential new marker, the authors should provide some evidence that miR-148b brings predictive prognostic information (i.e. T and S stages according to MEST-C histopathological score) after adjustment on basal GFR and albuminuria  (+/- blood pressure).

Reviewer 2 Report

  1.  

    Concerning reviewed paper I have some serious notes.

  2. Lack of a detailed description of the study group (comorbidity, age, pharmocological treatment etc) and the exclusion criteria.

  3. The description of the methods, results and discussion requires significant improvement. The majority of discussion consists of the results of that project.

  4. Conclusions do not flow directly from the presented data.

  5. There is no detailed description of the study limitation.

  6. The work does not bring any new clinical significant information in the field of nephrology.

  7. Language, used fonts, spaces between lines, description of tables as well as stylistic form should be carefully checked .

  8. The manuscript was not prepared exactly in accordance with the journal's requirements.

  1.